# The Effect of Magnetic Composites (γ-Al_2_O_3_/TiO_2_/γ-Fe_2_O_3_) as Ozone Catalysts in Wastewater Treatment

**DOI:** 10.3390/ma15238459

**Published:** 2022-11-28

**Authors:** Cheng Wang, Guangzhen Zhou, Yanhua Xu, Peng Yu

**Affiliations:** School of Environmental Science and Engineering, Nanjing Tech University, Nanjing 211816, China

**Keywords:** ozone catalysts, magnetic catalysts, advanced oxidation technology, wastewater treatment

## Abstract

Using municipal sewage as a source of reclaimed water is an important way to alleviate the shortage of water resources. At present, advanced oxidation technology (AOPs), represented by ozone oxidation, is widely used in wastewater treatment. In this study, γ-Al_2_O_3_, a low-cost traditional ozone catalyst, was selected as the matrix. By modifying magnetic γ-Fe_2_O_3_ with a titanate coupling agent, in situ deposition, and calcination, the final formation of a γ-Al_2_O_3_/TiO_2_/γ-Fe_2_O_3_ micrometer ozone catalyst was achieved. A variety of material characterization methods were used to demonstrate that the required material was successfully prepared. The catalyst powder particles have strong magnetic properties, form aggregates easily, and have good precipitation and separation properties. Subsequently, ibuprofen was used as the degradation substrate to investigate the ozone catalytic performance of the prepared catalyst, and this proved that it had good ozone catalytic activity. The degradation process was also analyzed. The results showed that in the ozone system, some of the ibuprofen molecules will be oxidized to form 1,4-propanal phenylacetic acid, which is then further oxidized to form 1,4-acetaldehyde benzoic acid and p-phenylacetaldehyde. Finally, the prepared catalyst was applied to the actual wastewater treatment process, and it also had good catalytic performance in this context. GC–MS detection of the water samples after treatment showed that the types of organic matter in the water were significantly reduced, among which nine pollutants with high content, such as bisphenol A and sulfamethoxazole, were not detected after treatment.

## 1. Introduction

At present, advanced oxidation technology (AOPs), represented by ozone oxidation, is widely used in wastewater treatment [1,2,3]. However, due to the low solubility and low utilization of ozone, the treatment efficiency of ozone oxidation technology is often not high [4,5]. Moreover, ozonation technology alone has a low complete mineralization rate for some refractory organic matter in water, and even produces more toxic intermediates, resulting in secondary pollution [6]. The use of catalysts can solve these problems to some extent. However, most of the catalysts widely reported in the literature require the use of precious metal nanomaterials for loading and other operations [7,8]. The industrial production process is relatively complex, and the cost is very high. Furthermore, little attention has been given to the secondary pollution caused by the leakage of precious metals.

Due to the surface properties and high specific surface area of γ-Al_2_O_3_, which has a wide range of sources and a low price, it is commonly used as a catalyst or catalyst support [9,10]. Many researchers have carried out ozone catalysis experiments with γ-Al_2_O_3_ alone. To further improve the catalytic effect, γ-Al_2_O_3_ has been used as a carrier to load some metal oxides, and its catalytic effect on ozone has been studied [11,12]. Among these, the active components generally include transition metals [13], precious metals [14], rare earth elements [15], and some metal oxides [16]. Certain precious metals have been shown to have the best effect, but it has also been found that the transition metals (Mn, Fe, Cu and Zn) [17], although not producing the best effect, still produce a good effect compared with many other metals, and transition metals are easily obtained [18]. Since many metal oxides can effectively decompose ozone in the gas and water phases, many researchers have considered the application of activated alumina-supported metal oxide catalysts in the field of wastewater treatment [19,20]. The improvement in catalytic ozonation efficiency is mainly due to the increase in hydroxyl radicals in the ozonation reaction [21]. In addition, whether in solution or on the surface of the catalyst, the deposited metal oxides can increase the number of hydroxyl groups on the surface of alumina and thereby improve the catalytic performance [22,23]. In conclusion, the application of γ-Al_2_O_3_-based catalysts in ozone catalytic systems can effectively improve the treatment efficiency of wastewater, which further proves the potential application potential of γ-Al_2_O_3_-based catalysts.

To further improve the efficiency of catalysts in ozone catalytic system, reducing catalyst particle size is an effective method [24]. However, the reduction in particle size brings another problem; that is, the sedimentation performance of the catalyst decreases, which leads to an increase in the catalyst loss rate with the water flow. Therefore, it is important to prepare a magnetic γ-Al_2_O_3_-based ozone catalyst with a small particle size. Effective deposition and separation can be achieved by using the magnetic properties of the catalyst itself. In recent years, magnetic core–shell structure materials have been widely used in the field of catalytic materials [25,26,27]. Among them, maghemite (γ-Fe_2_O_3_) is a kind of ferromagnetic cubic oxide, and because it is widely used in the production of magnetic materials and catalysts it is increasingly important in the field of material technology [28,29,30]. There are some studies on the catalytic oxidation of ozone. Magnetic Co and Mn-doped γ-Fe_2_O_3_ were prepared using the co-precipitation method and were found to be highly effective for the mineralization of 2,4-dichlorophenoxyacetic acid and its derivatives, 2,4-diclorophenol and 2,4,6-trichlorophenol, in aqueous solution with ozone [31]. A catalytic ozonation process using PAC/γ-Fe_2_O_3_ to effect Alizarin Red S degradation in aqueous solutions was studied [32].

In this study, γ-Al_2_O_3_, a low-cost traditional ozone catalyst, was selected as the matrix. According to the effect of the basic properties of γ-Al_2_O_3_ on the oxidation process, the material was improved, and a series of new catalysts were prepared. Through the modification of commercial hard magnetic γ-Fe_2_O_3_ using a titanate coupling agent, and in situ deposition, calcination finally led to the formation of a γ-Al_2_O_3_/TiO_2_/γ-Fe_2_O_3_ micron ozone catalyst. The catalysts were characterized using X-ray diffraction spectroscopy (XRD), scanning electron microscopy (SEM), X-ray photoelectron spectroscopy (XPS), Fourier infrared spectroscopy (FT-IR), a surface area test (BET), and isoelectric point (pH_zpc_) and surface hydroxyl group detection, and the physical and chemical properties of each series of materials were analyzed. Subsequently, ibuprofen was used as the degradation substrate to investigate the activity of the prepared catalyst. After determining the optimum ratio of catalyst, the catalyst with the best effect was selected to conduct an ozone catalytic experiment on actual wastewater. Finally, the biotoxicity and intermediates of degradation were investigated.

## 2. Materials and Methods

### 2.1. Materials and Instruments

Materials: Anhydrous aluminum chloride (AR, Sinopharm Chemical Reagent Co., Ltd., Shanghai, China); sodium hydroxide (AR, Sinopharm Chemical Reagent Co., Ltd.); ammonium hydroxide (AR, Sinopharm Chemical Reagent Co., Ltd.); hydrochloric acid (AR, Sinopharm Chemical Reagent Co., Ltd.); Isopropoxy trioleate oxy titanate (Nanjing Nengde New Material Co., Ltd., Nanjing, China); γ-Fe_2_O_3_ magnetic powder (Tangyin Zhongke Magneto Electric Co., Ltd., Nanjing, China) (the purity was greater than 99%).

Instruments: The crystalline phases of as-prepared catalysts were analyzed using XRD in the range of 2θ = 10–80° (X’TRA, Thermo Fisher Scientific, Walthamm, MA, USA). The spectra of the catalysts were recorded on an X-ray diffractometer with Cu Ka radiation (k = 1.5418 Å), and the scanning rate was 10° per min. The morphology and structure of the as-prepared samples were examined using SEM with an S-3400N II (Hitachi, Tokyo, Japan). The infrared spectra of the resins in the range of 4000–400 cm^−1^ were collected using a Nexus870 spectrophotometer (Nicolet, Thermo Fisher Scientific, Walthamm, MA, USA). The BET surface area and pore diameter of the catalysts were determined using an automatic analyzer (Micromeritics ASAP-2010C, Norcross, GA, USA) with N2 as the adsorbate.

### 2.2. Material Preparation

First, 1 g isopropoxy trioleate oxy titanate was dissolved in a triangle flask containing 100 mL methanol solution. Second, 2 g γ-Fe_2_O_3_ magnetic powder was added and stirred continuously (Tetrafluoroethylene stirring rod) at a slow speed (120 RPM) for 4 h. Next, 100 mL aluminum chloride solution (containing 2 g aluminum chloride) was poured into the mixture, and stirring was continued for 1 h. Ammonia was then quickly poured under stirring (80% of the theoretical amount), observing the rotation of the stirring rod. If the solution becomes sticky, a little ammonia should be added, the glass rod should be dipped into the solution, and drops of the solution placed onto the pH test paper to determine whether the pH is between 8 and 9. If it is, no further ammonia need be added, and stirring should continue. The pH should be measured regularly, and if there is a drop then ammonia should be added. After stirring for 30 min, the liquid was allowed to stand for 1 h. It was then poured into the suction filter funnel to filter and washed continuously with deionized water until the liquid was neutral. After the filter cake was taken out, it was put into the drying oven to dry at 60 °C for 12 h, and finally put it into the Muff furnace for calcination at 550 °C for 4 h at high temperature to get 1:0.5:1 γ-Al_2_O_3_/TiO_2_/γ-Fe_2_O_3_ (the ratio here is the mass of initial aluminum chloride/mass of isopropoxytrioleate oxy-titanate/mass of γ-Fe_2_O_3_ magnetic powder). Other proportions followed a similar approach.

### 2.3. Analysis Method

Surface hydroxyl group determination: 0.3 g catalyst particles were added to a series of NaOH solutions (50 mL each, concentration range 2 mM to 100 mM). After shaking at 25 °C for more than 4 h, the solution was filtered through a 0.45 μm membrane. The supernatant was titrated with a standard HNO_3_ solution to determine the residual NaOH. Since the acidic hydroxyl group reacts with NaOH, the density could be quantified by NaOH consumption. Based on the principle of charge balance, acidic and basic hydroxyl groups were quantified equally. As a result, the total density of the hydroxyl groups on the surface was twice that of the acid.

Catalyst isoelectric point (pH_zpc_) determination: 25 mL of 0.01 M NaCl solution was added to a series of 50 mL capped glass tubes. The initial pH of the solution was adjusted to between 2 and 11 by using 0.1 M HCl or NaOH. In addition, 0.05 g catalyst particles were added to each tube, and the suspension was then stirred in an oscillator at 150 rpm at room temperature. After 24 h, the suspension was filtered, and the final pH of the solution was determined. The difference between the initial and final pH relative to the initial pH was plotted. The intersection point of the obtained curve and the abscissa was the isoelectric point value.

In a typical catalytic ozonation procedure, 0.5 g catalyst is mixed with 500 mL biological effluents in a flask under magnetic stirring. The flowing rate of O_3_ during the ozonation process is kept at 0.5 L/min, and the concentration is 4.0 mg/L. During the reaction process, 5 mL solution is taken out from the reactor for analysis at 2 min intervals. A similar experimental process was carried out using only ozone. Then, the filtrates were analyzed by performance liquid chromatography (HPLC). Determination of chemical oxygen demand in accordance with the national standard method GB11914-89: UV absorbance at 254 nm (UV_254_) was measured on a Shimadzu UV-2100 spectrophotometer. After each degradation test, the catalyst was removed and dried for 48 h in an oven at 50 °C. The final catalyst mass was then weighed and poured into the solution according to the amount of catalyst left. The catalyst dosage was kept at 1 g/L.

Toxicity evaluation: Before the test, the pH of the sample solution was adjusted to 8.0, and a bioluminescence test was performed according to the ISO standard method. After activation and incubation, the bacteria were exposed to the reaction solution at 20 °C for 15 min, and the bioluminescence was measured on a biotoxicity tester (DLY-3). The inhibition rate of each sample against the blank control was then calculated. To ensure the accuracy of the test, all sample tests and control experiments must be performed.

## 3. Results and Discussion

### 3.1. Material Characterization

The XRD analysis of the magnetic γ-Al_2_O_3_-based ozone catalyst prepared with γ-Fe_2_O_3_ as a magnetic core was carried out, and the specific results are shown in Figure 1. As can be seen from Figure 1, the peak shapes of the three γ-Al_2_O_3_/TiO_2_/γ-Fe_2_O_3_ samples prepared are basically the same: the peak shapes are sharp and the half-peak width is narrow, indicating that the crystallinity of the samples is good. Among them, the main characteristic peak and cubic crystal phase of the γ-Al_2_O_3_ was significant (standard card JCPDS 50-0741) [33]. It was found that when 2θ was 37°, 46.9°, and 67°, the characteristic peaks corresponding to crystal plane, (311), (400), and (440), appeared. By comparing the prepared composite catalysts, it was found that they also appeared to have an obvious characteristic peak at about 67°, corresponding to the characteristic peak of the XRD pattern of the γ-Al_2_O_3_. For the pure γ-Fe_2_O_3_ magnetic powder, the main characteristic peaks were compared with the cubic crystal phase (standard card JCPDS 25-1402) [34]. The diffraction peaks at 2θ were 30.5°, 35.8°, 43.5°, 53.7°, 57.5°, and 63.3°, corresponding to crystal planes of (220), (311), (440), (422), (511), and (440), respectively. It was also found that the XRD patterns of the γ-Al_2_O_3_/TiO_2_/γ-Fe_2_O_3_ samples also had these characteristic diffraction peaks, and there was a certain amount of γ-Fe_2_O_3_ on the surface. As for the XRD pattern of the pure TiO_2_, it was found that the diffraction peaks at 2θ were 25.2°, 37.5°, 47.8°, 54.2°, and 63.5°, corresponding to planes of (101), (112), (200), (211), and (204), respectively. The corresponding standard card is JCPDS 21-1272 [35]. In summary, the XRD patterns of the prepared magnetic γ-Al_2_O_3_/TiO_2_/γ-Fe_2_O_3_ samples had the characteristic diffraction peaks of γ-Al_2_O_3_, TiO_2_, and γ-Fe_2_O_3_, which means that the three successfully formed a composite catalyst. Further observation of the map showed that with the further increase in the amount of the titanate coupling agent, the characteristic peak intensity of γ-Al_2_O_3_ near 67° was weakened, which meant that the content of γ-Al_2_O_3_ decreased.

XPS is one of the important means of analyzing the valence of elements on the surface of substances. Therefore, XPS analysis and tests were carried out on the prepared magnetic catalysts γ-Al_2_O_3_/TiO_2_/γ-Fe_2_O_3_ with different ratios, as shown in Figure 2. Figure 2A shows the Al 2p high-resolution energy spectrum of the sample. The electron binding energy is about 73.6 eV, which is mainly caused by the characteristic peak of the Al3+ in the γ-Al_2_O_3_ [36]. Figure 2B shows the energy spectrum of the Ti 2p of the sample, and it can be seen that there are two characteristic peaks, the electron binding energies of which are 457.5 eV and 463.4 eV, which are a result of the Ti in the +4 valence state and a small amount of Ti in the +3 valence state of the TiO_2_ in the preparation catalyst [37]. The high-resolution spectrum of the Fe 2p is shown in Figure 2C. The two characteristic peaks of the Fe 2p in the γ-Fe_2_O_3_ are located at 710.91 eV and 723.34 eV, indicating that the Fe element mainly exists as Fe^3+^ in the sample [38]. Figure 2D shows the high-resolution energy spectrum of the O 1s. According to the peak software (XPS peak fit v4.1), the O 1s has three characteristic peaks at 529.5 eV, 531.7 eV, and 532.5 eV, which correspond to oxygen in the lattice of the γ-Fe_2_O_3_ (Fe-O), oxygen in the outer adsorbed water (H_2_O), and hydroxyl oxygen on the surface of the γ-Al_2_O_3_ (O-H). Through the above XPS characterization analysis, it can be seen that the prepared magnetic γ-Al_2_O_3_-based ozone catalyst contains γ-Al_2_O_3_, TiO_2_, and γ-Fe_2_O_3_, which further indicates that the prepared ozone catalyst is a γ-Al_2_O_3_/TiO_2_/γ-Fe_2_O_3_ composite catalytic material.

In order to further understand the differences in the external morphology of the γ-Al_2_O_3_/TiO_2_/γ-Fe_2_O_3_ catalysts prepared using different proportions, scanning electron microscopy (SEM) analysis was carried out. Figure 3 shows the SEM images of the γ-Al_2_O_3_/TiO_2_/γ-Fe_2_O_3_ (1:0.5:1), γ-Al_2_O_3_/TiO_2_/γ-Fe_2_O_3_ (1:1:1), and γ-Al_2_O_3_/TiO_2_/γ-Fe_2_O_3_ (1:2:1). It can be seen that the external morphologies of the catalysts prepared using different proportions of raw materials have certain differences. As shown in Figure 3A, the γ-Al_2_O_3_/TiO_2_/γ-Fe_2_O_3_ (1:0.5:1) mainly presents a needle-like shape, with a few agglomerations and basically a uniform appearance. Figure 3B shows the γ-Al_2_O_3_/TiO_2_/γ-Fe_2_O_3_ (1:1:1) catalyst. It can be seen that there is an increase in clustering, but that the overall arrangement of the clusters together into large particles is reduced. However, in Figure 3C, they are basically clustered together, making the particles larger, which may affect the ability of the particles in the treatment of pollutants. In addition, the EDS analysis embedded in Figure 3C shows that there are Al, Fe, O, and Ti elements in the material, which further indicates that the magnetic γ-Al_2_O_3_/TiO_2_/γ-Fe_2_O_3_ catalyst has been successfully prepared.

Similarly, FT-IR spectra were used to further understand the physical and chemical characteristics of the prepared catalyst γ-Al_2_O_3_/TiO_2_/γ-Fe_2_O_3_ (1:1:1), as shown in Figure 4. As can be seen from the figure, for the γ-Al_2_O_3_ monomer, the stretching vibration peaks generated by the 3100 cm^−1^, 1380 cm^−1^, 1050 cm^−1^, and 710 cm^−1^ infrared bands can be attributed to the O-H, Al-O, Al-O-Al, and O-Al-O, respectively [39]. For the TiO_2_ monomer, the vibration peak of 1320 cm^−1^ can be attributed to the stretching vibration of the Ti-O [40]. For the γ-Fe_2_O_3_ monomer, the characteristic peaks of 500–750 cm^−1^ can be attributed to the stretching vibration of the Fe-O [41]. Finally, by observing the γ-Al_2_O_3_/TiO_2_/γ-Fe_2_O_3_ catalysts we prepared, we found several characteristic peaks. This further indicates that the prepared catalysts contain these three kinds of metal oxides, thus proving that the magnetic catalyst γ-Al_2_O_3_/TiO_2_/γ-Fe_2_O_3_ was successfully prepared.

For the magnetic γ-Al_2_O_3_-based ozone catalysts γ-Al_2_O_3_/TiO_2_/γ-Fe_2_O_3_ (1:0.5:1), γ-Al_2_O_3_/TiO_2_/γ-Fe_2_O_3_ (1:1:1), and γ-Al_2_O_3_/TiO_2_/γ-Fe_2_O_3_ (1:2:1), other physical and chemical properties are shown in Table 1. The specific surface areas were 203.15 m^2^/g, 210.23 m^2^/g, and 152.71 m^2^/g, and the amounts of hydroxyl groups on the surface were 0.32 mmol/g, 0.36 mmol/g, and 0.27 mmol/g, respectively. The specific surface area of the γ-Al_2_O_3_/TiO_2_/γ-Fe_2_O_3_ (1:2:1) was significantly lower than that of the same series of catalysts prepared using the other two ratios, and the corresponding amount of surface hydroxyl was also less than the other two catalysts.

Vibrating sample magnetometry (VSM) is a widely preferred method for measuring and characterizing the magnetic properties of nanoparticles, such as magnetization, retentivity, and coercivity. Appendix A shows the magnetic properties of the prepared catalyst, which prove that the prepared catalyst has the required magnetic properties, making it suitable for reuse after being collected from water.

### 3.2. Material Catalytic Activity

Subsequently, ibuprofen was used as the degradation substrate to investigate the activity of the prepared catalyst. As shown in Figure 5, the degradation efficiency of the magnetic γ-Al_2_O_3_ ozone catalyst for ibuprofen was demonstrated. It can be seen that both ozone alone and γ-Al_2_O_3_ as catalyst in the solution, under the ozone system, showed a certain ability to degrade ibuprofen, with 10-min degradation rates of about 30% and 37%, respectively. In the system of the magnetic γ-Al_2_O_3_-based ozone catalyst prepared using three different ratio methods, it can be seen that the γ-Al_2_O_3_/TiO_2_/γ-Fe_2_O_3_ (1:0.5:1), γ-Al_2_O_3_/TiO_2_/γ-Fe_2_O_3_ (1:1:1), and γ-Al_2_O_3_/TiO_2_/γ-Fe_2_O_3_ (1:2:1) can degrade about 45%, 65%, and 44% of the ibuprofen, respectively, within 10 min. Compared with the ozone system alone, this was a significant increase of 15.35% and 14%. Even compared with the pure alumina ozonation system, it was improved by 8.28% and 7%. This indicates that the formation of the composite catalytic materials can effectively improve the catalytic efficiency of ozone. It can also be seen that the ozone catalytic efficiency of the γ-Al_2_O_3_/TiO_2_/γ-Fe_2_O_3_ (1:1:1) was significantly better than that of the γ-Al_2_O_3_/TiO_2_/γ-Fe_2_O_3_ (1:0.5:1) and the γ-Al_2_O_3_/TiO_2_/γ-Fe_2_O_3_ (1:2:1). This is also the result of using different ratio in their preparation.

The study of the degradation kinetics in the catalytic process is helpful for the regulation of various process parameters in the practical application of catalysts. Therefore, the degradation kinetics of ozone alone, γ-Al_2_O_3_, γ-Al_2_O_3_/TiO_2_/γ-Fe_2_O_3_ (1:0.5:1), γ-Al_2_O_3_/TiO_2_/γ-Fe_2_O_3_ (1:1:1), and γ-Al_2_O_3_/TiO_2_/γ-Fe_2_O_3_ (1:2:1) in several ozone systems were analyzed, as shown in Figure 6. The specific parameters are shown in Table 2. It can be seen that the fitted first-order kinetic k values for the ozone alone, γ-Al_2_O_3_, γ-Al_2_O_3_/TiO_2_/γ-Fe_2_O_3_ (1:0.5:1), γ-Al_2_O_3_/TiO_2_/γ-Fe_2_O_3_ (1:1:1), and γ-Al_2_O_3_/TiO_2_/γ-Fe_2_O_3_ (1:2:1) ozone systems were 0.040 min^−1^, 0.047 min^−1^, 0.061 min^−1^, 0.105 min^−1^, and 0.062 min^−1^, respectively. When γ-Al_2_O_3_/TiO_2_/γ-Fe_2_O_3_ (1:1:1) was used as the catalyst, the k value was about 2.5 times that of ozone alone; that is, the reaction rate was 2.5 times faster than that of ozone alone. Even when compared with the γ-Al_2_O_3_ reaction system, the reaction rate was about 2 times faster. The curve deviation R^2^ of the fitted degradation curves of the ozone systems were 0.996, 0.993, 0.997, 0.992, and 0.992, respectively, showing good correlation, which means that it is feasible to use it as a first-order kinetic model to simulate the degradation effectiveness when used as a catalyst.

In order to evaluate whether some intermediates are produced in the process of ozone oxidation and ozone catalytic oxidation, which may cause a potential risk of secondary pollution to the aquatic environment, luminescent bacteria toxicity was used to evaluate the toxicity of the solution before and after the reaction, as shown in Figure 7. As shown in Figure 7, the inhibition rate of the original ibuprofen solution against luminescent bacteria was about 50%, and the inhibition rate decreased to about 38% after ozone oxidation alone. After adding the γ-Al_2_O_3_/TiO_2_/γ-Fe_2_O_3_ (1:0.5:1) catalyst, the toxicity was further reduced, and the inhibition rate was reduced to about 31%. When the γ-Al_2_O_3_/TiO_2_/γ-Fe_2_O_3_ (1:1:1) catalyst was added, the inhibition rate of the luminescent bacteria was further reduced to about 20% compared with the system of the γ-Al_2_O_3_/TiO_2_/γ-Fe_2_O_3_ (1:0.5:1) catalyst. After adding the γ-Al_2_O_3_/TiO_2_/γ-Fe_2_O_3_ (1:0.5:1) catalyst, although the toxicity also decreased, it was still higher compared with that achieved using the γ-Al_2_O_3_/TiO_2_/γ-Fe_2_O_3_ (1:1:1) catalyst. This shows that the series of catalysts we prepared can play a certain role in reducing toxicity and will not cause secondary pollution of the water environment. It should be noted that the toxicity reduction effects of the catalysts with different preparation ratios are not the same.

In order to further understand which intermediates produced by ozonation alone may cause secondary pollution to the water environment, liquid mass spectrometry was used to test the solution of ibuprofen oxidized by ozonation alone, and the results are shown in Figure 8. Figure 8A shows the mass-to-charge ratio of ibuprofen, and its *m*/*z* value is 205.04 ([M-H]-). After ozonation treatment, the main three substances were analyzed, as shown in Figure 8B,C. Their mass-to-charge ratios are 219.28 ([M-H]-), 163.04 ([M-H]-), and 161.23 ([M-H]-), respectively. The substances that may be formed are 1,4-propanal phenylacetic acid, 1,4-acetaldehyde benzoic acid, and p-phenylacetaldehyde. It can be concluded that in the system using ozone alone, some of the ibuprofen molecules would be oxidized to form 1,4-propanal phenylacetic acid, which would then be further oxidized to form 1,4-acetaldehyde benzoic acid and p-phenylacetaldehyde. The formation of these intermediates may lead to changes in toxicity in the water. Possible degradation paths are shown in Appendix A.

### 3.3. Effect on the Treatment of Actual Wastewater

Many studies use only catalysts in a single system for catalyst evaluation, ignoring the actual research on wastewater purification. Therefore, we carried out ozone catalytic experiments on real wastewater using the prepared catalyst.

Chemical oxygen demand (COD) is one of the most important indicators for the actual wastewater removal effect. The prepared magnetic γ-Al_2_O_3_ catalyst was used to conduct ozone catalytic experiments on actual wastewater, and the specific results are shown in Appendix A. It can be seen from Appendix A that the COD removal rate after ozone oxidation alone was only about 22%. When the γ-Al_2_O_3_/TiO_2_/γ-Fe_2_O_3_ series catalysts are introduced, the COD removal rate reached about 40–55% through ozone catalysis. It can be concluded that magnetic γ-Al_2_O_3_ catalysts can remove COD from actual wastewater, and that γ-Al_2_O_3_/TiO_2_/γ-Fe_2_O_3_ (1:1:1) has the best catalytic effect. UV_254_ is the absorbance of some organic matter in water under 254 nm UV light, which reflects the amount of humus macromolecular organic matter and aromatic compounds containing double bond C=C and double bond C=O naturally existing in water. Similarly, the prepared magnetic γ-Al_2_O_3_ catalyst was used to conduct ozone catalytic experiments on actual wastewater, and the specific results are shown in Appendix A. It can be seen from Appendix A that the removal rate of UV_254_ after ozonation alone was only about 60%. When the γ-Al_2_O_3_/TiO_2_/γ-Fe_2_O_3_ series catalysts were introduced, the COD removal rate reached about 70–80% through ozone catalysis. It can also be concluded that the magnetic γ-Al_2_O_3_ catalysts can remove COD in actual wastewater, and that γ-Al_2_O_3_/TiO_2_/γ-Fe_2_O_3_ (1:1:1) has the best catalytic effect.

In practical applications, the stability of catalysts is an important index to investigate. Therefore, the results of the cycle experiment on the catalyst are shown in Appendix A. It can be seen that after 10 experiments, the COD removal efficiency was reduced from 55% to about 47%. It was also found that the degradation efficiency decreased rapidly after the first few experiments, but did not decrease significantly after the final five experiments. This may be because some of the active components on the surface of the catalyst are not closely connected, and some of them fall off in the early stage, which leads to a significant decline in the initial catalytic efficiency. The crystal shape of the catalyst before and after the reaction was characterized using XRD, as shown in Appendix A. According to Appendix A, the crystal structure of the magnetic γ-Al_2_O_3_/TiO_2_/γ-Fe_2_O_3_ catalyst is also relatively stable and does not change significantly. This also explains the relatively stable performance of the catalyst.

In order to further understand the process both before and after the ozone catalytic effect, a GC–MS analysis of the water quality was used to determine the specific materials in the water that were removed or reduced after the ozone catalytic oxidation, as well as those that were increased. The specific main organic analysis is shown in Appendix A, and a gas chromatograph chart is shown in Figure 9. It can be seen from Figure 9A that several obvious characteristic peaks appear in the water samples before treatment at around 8–14 min. However, in the analysis of water quality after treatment (Figure 9B), several characteristic peaks appear in the water samples before 8–14 min, obviously disappear, or decrease, which indicates that a variety of substances have been removed, or partially removed, from the water. Similarly, for the peak effect around 18–28 min after the GC–MS analysis of the water quality before treatment, many characteristic peaks also disappeared or decreased. Therefore, we speculate that a variety of organic pollutants in water may be removed or decomposed after an ozone catalytic reaction. According to the results in Appendix A, a dozen substances with a relatively high peak area detected after treatment were selected for classification. It can be seen that n-butaisoctyl phthalate, sulfamethoxazole, bisphenol A, and other macromolecular substances were oxidized and decomposed into small molecular organic matter, and that some aromatic substances containing benzene rings were also easy degraded. On the other hand, a considerable number of alkanes, such as 2,6,11-trimethylundecane and 3-methyl-5-propyl nonane, were detected after the reaction. Some of these substances may be obtained from the decomposition of macromolecular organic matter, and some may be obtained from the decomposition of small molecular organic matter and then re-paired to form a new substance. In general, alkanes and small molecules are the main substances in water after ozone-catalyzed reactions.

## 4. Conclusions

In this study, γ-Al_2_O_3_/TiO_2_/γ-Fe_2_O_3_ was successfully synthesized by modifying magnetic γ-Fe_2_O_3_ with a titanate coupling agent followed by in situ deposition and calcination. Subsequently, a series of characterizations (XRD, SEM, XPS, and FT-IR) proved that the prepared catalyst was magnetic γ-Al_2_O_3_/TiO_2_/γ-Fe_2_O_3_. Ibuprofen was then used as the degradation substrate to investigate the activity of the prepared catalyst. This investigation indicated that the formation of composite catalytic materials can effectively improve the catalytic efficiency of ozone. The ozone catalytic efficiency of γ-Al_2_O_3_/TiO_2_/γ-Fe_2_O_3_ (1:1:1) was significantly better than that of γ-Al_2_O_3_/TiO_2_/γ-Fe_2_O_3_ (1:0.5:1) and γ-Al_2_O_3_/TiO_2_/γ-Fe_2_O_3_ (1:2:1). To further understand the degradation process of ibuprofen in ozonation systems, liquid mass spectrometry was used to test a solution of ibuprofen oxidized by ozonation. The results showed that in the ozone system, some of the ibuprofen molecules were be oxidized to form 1,4-propanal phenylacetic acid, which was then further oxidized to form 1,4-acetaldehyde benzoic acid and p-phenylacetaldehyde. The formation of these intermediates may lead to changes in toxicity in the water. In addition, ozone catalytic experiments were carried out on real wastewater using the prepared catalyst. It can be concluded that magnetic γ-Al_2_O_3_ catalysts can remove COD in actual wastewater, and that γ-Al_2_O_3_/TiO_2_/γ-Fe_2_O_3_ (1:1:1) has the best catalytic effect. Finally, in order to further understand the process before and after the ozone catalytic effect, a GC–MS analysis was performed to determine water quality. The GC–MS analysis of the water samples after treatment showed that the types of organic matter in the water were significantly reduced, among which nine pollutants with high content, such as bisphenol A and sulfamethoxazole, were not detected after treatment.

## Figures and Tables

**Figure 1 materials-15-08459-f001:**
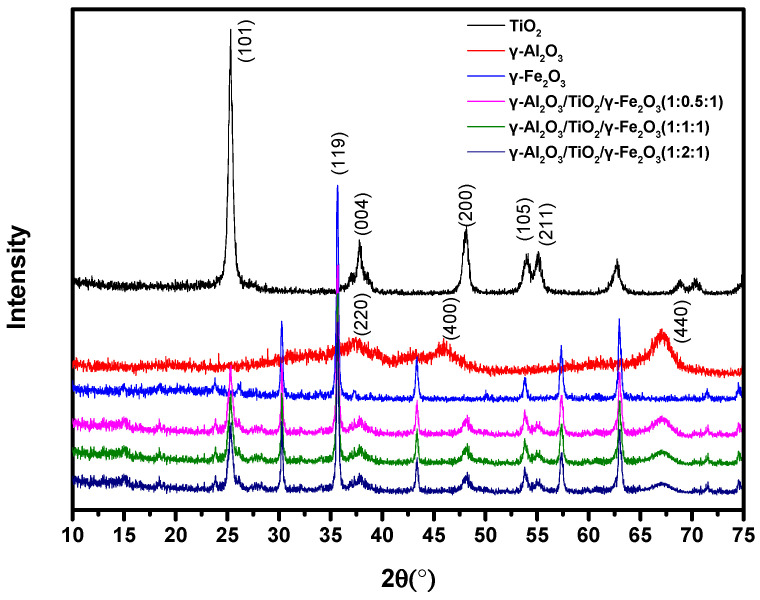
XRD spectra of γ-Al_2_O_3_/TiO_2_/γ-Fe_2_O_3_.

**Figure 2 materials-15-08459-f002:**
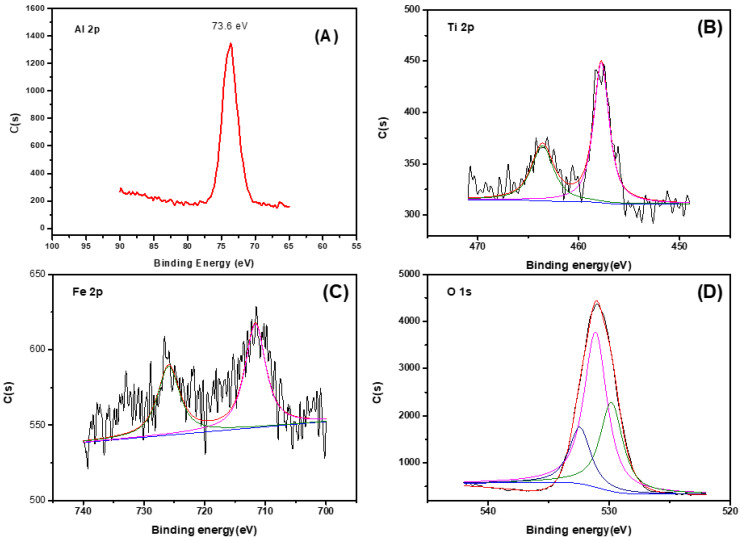
XPS patterns of γ-Al_2_O3/TiO_2_/γ-Fe_2_O_3_: (**A**) Fe 2p; (**B**) Al 2p; (**C**) Fe 2p; (**D**) O 1s.

**Figure 3 materials-15-08459-f003:**
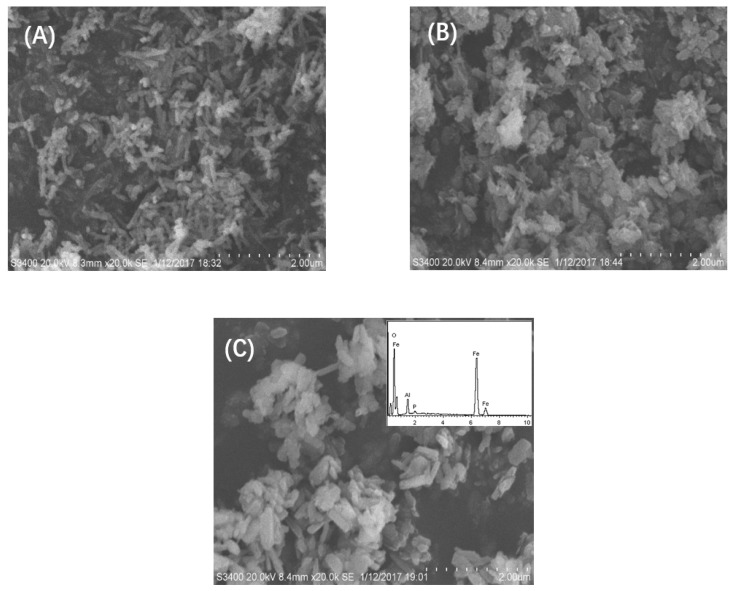
SEM images of catalysts prepared using different proportions: (**A**) γ-Al_2_O_3_/TiO_2_/γ-Fe_2_O_3_ (1:0.5:1); (**B**) γ-Al_2_O_3_/TiO_2_/γ-Fe_2_O_3_ (1:1:1); (**C**) γ-Al_2_O_3_/TiO_2_/γ-Fe_2_O_3_ (1:2:1).

**Figure 4 materials-15-08459-f004:**
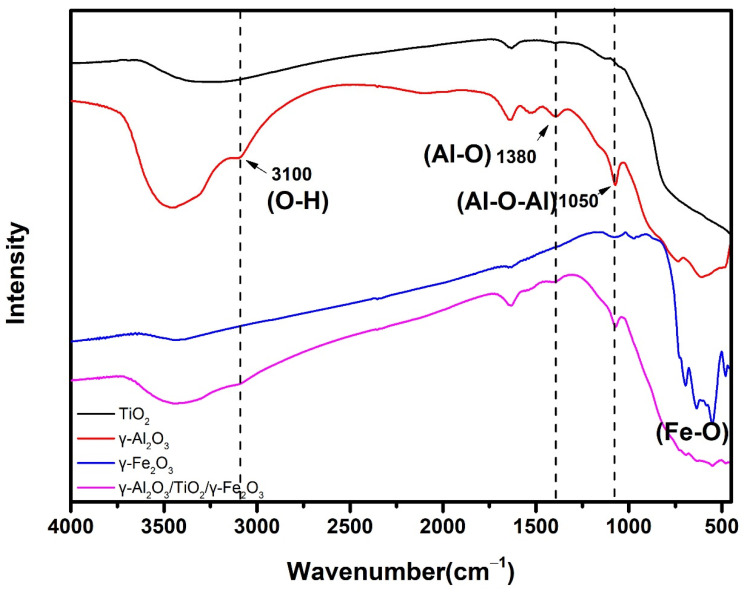
FT-IR spectra of γ-Al_2_O_3_/TiO_2_/γ-Fe_2_O_3_, TiO_2_, γ-Fe_2_O_3_, and γ-Al_2_O_3_.

**Figure 5 materials-15-08459-f005:**
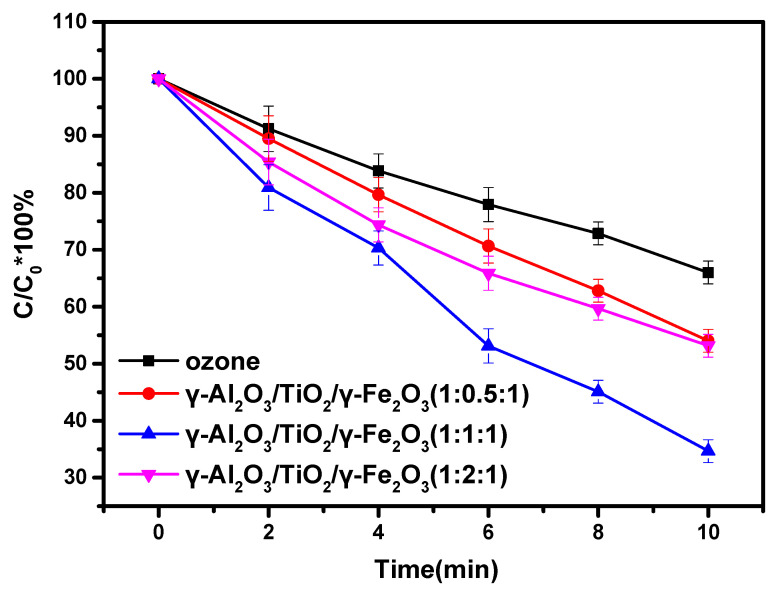
Degradation of ibuprofen by magnetic γ-Al_2_O_3_-based ozone catalyst.

**Figure 6 materials-15-08459-f006:**
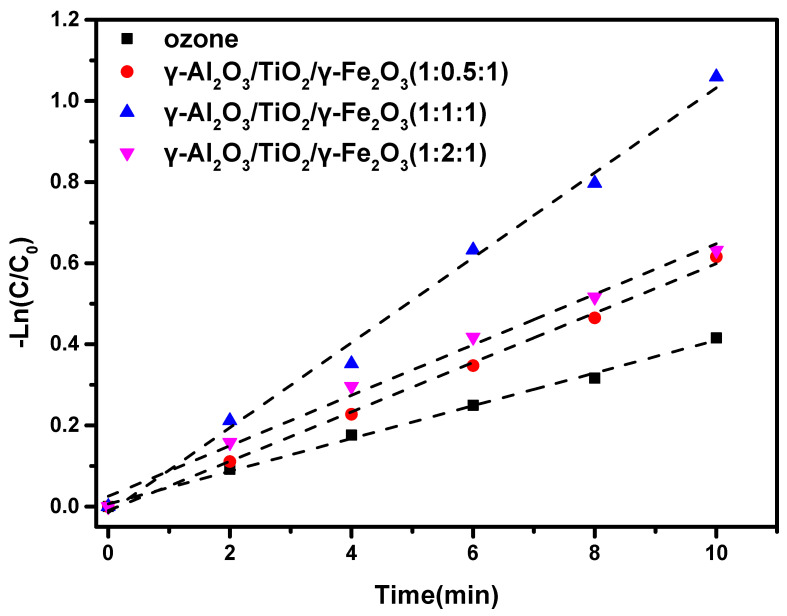
Kinetic study of ibuprofen degradation by magnetic γ-Al_2_O_3_ base ozone catalyst.

**Figure 7 materials-15-08459-f007:**
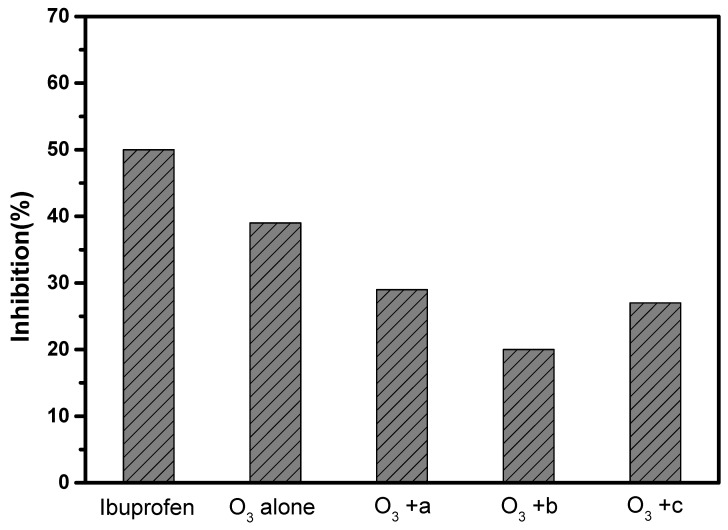
Toxicity analysis of catalysts before and after degradation of ibuprofen: (a) γ-Al_2_O_3_/TiO_2_/γ-Fe_2_O_3_ (1:0.5:1); (b) γ-Al_2_O_3_/TiO_2_/γ-Fe_2_O_3_ (1:1:1); (c) γ-Al_2_O_3_/TiO_2_/γ-Fe_2_O_3_ (1:2:1).

**Figure 8 materials-15-08459-f008:**
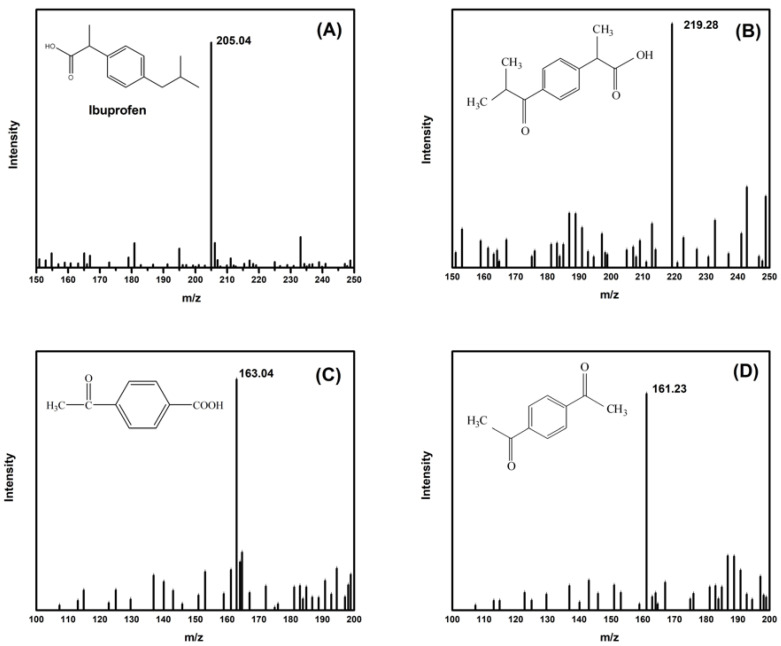
A possible intermediate of ibuprofen after ozonation. (**A**) ibuprofen; (**B**) 1,4-propanal phenylacetic acid; (**C**) 1,4-acetaldehyde benzoic acid; (**D**) p-phenylacetaldehyde.

**Figure 9 materials-15-08459-f009:**
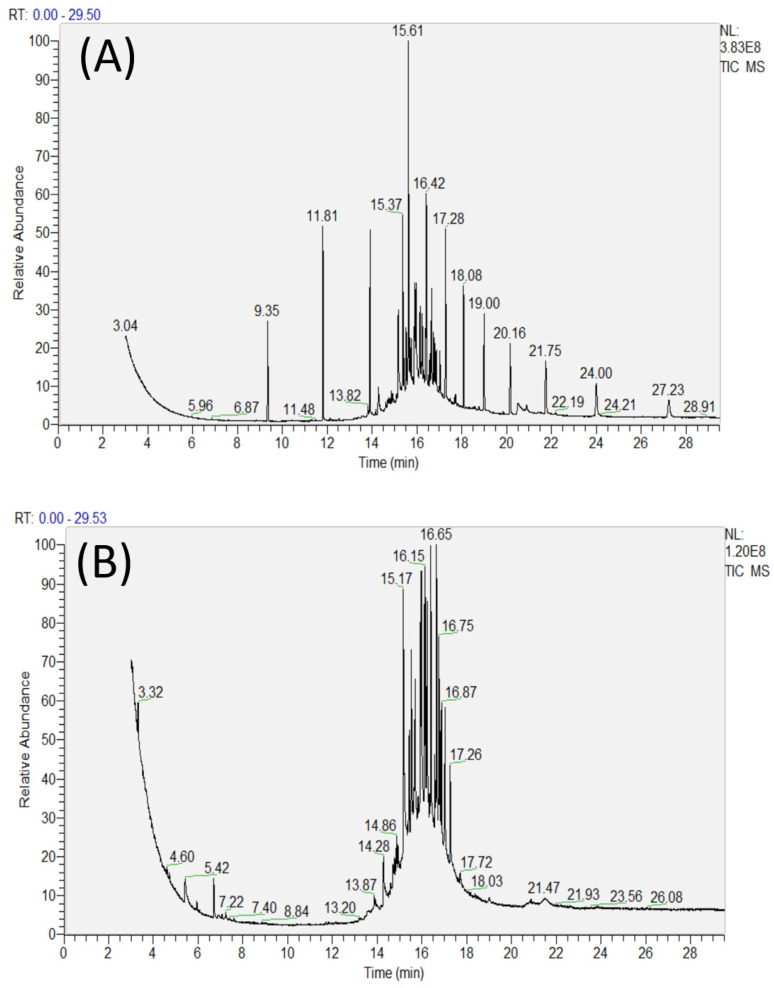
Gas chromatograms of water samples before (**A**) and after (**B**) ozone catalytic treatment.

**Table 1 materials-15-08459-t001:** Specific surface area, pore volume, pore size, surface hydroxyl, and pH_zpc_ of the catalyst.

Catalyst	Specific Surface Area (m^2^/g)	Pore Volume (cm^3^/g)	Surface Hydroxyl (mmol/g)	pH_zpc_
γ-Al_2_O_3_/TiO_2_/γ-Fe_2_O_3_(1:0.5:1)	203.15	0.23	0.32	7.21
γ-Al_2_O_3_/TiO_2_/γ-Fe_2_O_3_(1:1:1)	210.23	0.21	0.36	7.16
γ-Al_2_O_3_/TiO_2_/γ-Fe_2_O_3_(1:2:1)	152.71	0.19	0.27	7.17

**Table 2 materials-15-08459-t002:** First-order dynamics parameters (magnetic γ-Al_2_O_3_-based ozone catalyst).

Series	Catalyst	First Order Kinetic Equation	*k* (min^−1^)	*R* ^2^
1	Ozone	−ln(C/C_0_) = 0.040 t	0.040	0.996
2	γ-Al_2_O_3_/TiO_2_/γ-Fe_2_O_3_(1:0.5:1)	−ln(C/C_0_) = 0.061 t	0.061	0.997
3	γ-Al_2_O_3_/TiO_2_/γ-Fe_2_O_3_(1:1:1)	−ln(C/C_0_) = 0.105 t	0.105	0.992
4	γ-Al_2_O_3_/TiO_2_/γ-Fe_2_O_3_(1:2:1)	−ln(C/C_0_) = 0.062 t	0.062	0.992

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
