# Peer review of "The Effect of Magnetic Composites (γ-Al2O3/TiO2/γ-Fe2O3) as Ozone Catalysts in Wastewater Treatment"

_materials, 2022, doi:10.3390/ma15238459_

Round 1

Reviewer 1 Report

The current work focuses on the effect of magnetic composites (γ-Al2O3/TiO2/γ-Fe2O3) as ozone catalysts in wastewater treatment. The author’s some effort into the manuscript, but major issues should be addressed. Extensive editing of English language and style required.

Abstract

-Abstract should be rephrased with the main novelty and main outputs not general

- Correct typo, γ-Al2O3/TiO2/γ-Fe2O3

Introduction

-  The introduction provides sufficient background, and all relevant references are included.

- Correct typo, Fe2O3

Materials and methods section 

- The impurities of the used materials should be inserted

- What is the condition of XRD analysis? Scanning rate?

- -Correct typo, Fe2O3, cm-1, N2,…

- Stirred continuously! What type of used stirrer?

- Then pour 100 ml aluminum chloride solution! One shot or drop by drop?

- What is the specific amount of used ammonia?

- Please more details in experimental methods are required for easier reproducing by the reader

- 2.2 Material preparation, please correct the starting and ending sentences of this section

- In the experimental section, no information about the cycling of COD removal

Results and discussion

- Please insert the indexed peaks on the XRD figure for easier comparison by the reader

- Please calculate the size from XRD analysis and compare it with the size obtained from SEM analysis

- Please insert the target groups with their positions on the IR figure for easier comparing

- The efficiency of the prepared catalyst depends mainly on the elemental content. Please elemental analysis is required to evaluate this point

- VSM is required to investigate the magnetic properties of the prepared magnetic catalysts

- Error par should be inserted in figures

- Fig. S2 Catalyst cycle experiment (A), how at cycle 10 more efficient than cycle 4?? 

Author Response

The uploaded file.

Reviewer 2 Report

Comments and Suggestions for Authors

Manuscript ID: materials-2041176

Type: Article

Title: The effect of magnetic composites (γ-Al2O3/TiO2/γ-Fe2O3) as ozone catalyst in wastewater treatment

Authors: Cheng Wang, Guangzhen Zhou, Yanhua Xu*, Peng Yu

Comments: The proposed work demonstrates the effect of magnetic composites (γ-Al2O3/ TiO2/ γ-Fe2O3) as ozone catalyst in wastewater treatment. It requires a major correction since the work done is fine. Authors are required to address these comments for the improvement of the paper.

·         The author writes a novelty statement about their research. needs to express the overall scientific outcome of the proposed work. The introduction is weak. include more appropriate recent research works.

·         The author made many typos. For example, when writing chemical formulas, authors have to carefully check whether the written formulae are correct or not. Please check and correct the given formula (Ex: corrected is γ-Al2O3/TiO2/γ-Fe2O3) Use the MDPI template while revising the manuscript. Maintain the same fonts, spacing, references style and place figures and tables near to the cited text.

·         Why has the author not supported their results with previous literature? what was the λ (lambda) value used in the XRD calculations

·         Include a table that makes a comparison with other works of a similar nature that have been reported.

·         Authors must include a flow chart for methodology and a list of abbreviations in the last section of the revised MS.

·         Conclusions should be more precise rather than discussion.

Author Response

 The uploaded file

Round 2

Reviewer 1 Report

The author’s great effort into the manuscript, but major minors should be addressed.

- The authors answer all my previous comments but it remains two comments were not answered:

Please calculate the size from XRD analysis and compare it with the size obtained from SEM analysis

The efficiency of the prepared catalyst depends mainly on the elemental content. Please elemental analysis is required to evaluate this point

-          The title of the manuscript in the supplementary file is “Nitrogen doped cobalt anchored on the used resin-based catalyst to activate peroxymonosulfate for the degradation of ibuprofen” not as the title in the main file “The effect of magnetic composites (γ-Al2O3/TiO2/γ-Fe2O3) as ozone catalysts in wastewater treatment” which one is the right?

-          Line 203, 222,223,238 correct typo

-          Vibrating Sample Magnetometry (VSM) analysis should be inserted in experimental work with the model

-          Reference 14, correct typo

-          VSM analysis, Zoom in figure around zero to clear the supermagnetic behavior

-          Line 253, “due to the high saturation magnetization value” why you claim high saturation magnetization value, it is only around 2 emu/g!!

Author Response

Thanks for the reviewer’s careful work.

Reviewer 2 Report

The revised manuscript can be accepted 

Author Response

Thanks for the reviewer’s kind advice and recognition